# The Rabep1-Mediated Endocytosis and Activation of Trypsinogen to Promote Pancreatic Stellate Cell Activation

**DOI:** 10.3390/biom12081063

**Published:** 2022-07-31

**Authors:** Wenchao Yao, Dankun Luo, Zhenyi Lv, Yang Yang, Liyi Wang, Biao Ma, Dongbo Xue, Chenjun Hao, Yingmei Zhang

**Affiliations:** 1Key Laboratory of Hepatosplenic Surgery, Ministry of Education, The First Affiliated Hospital of Harbin Medical University, Harbin 150001, China; yaowenchao@hrbmu.edu.cn (W.Y.); luodankun@hrbmu.edu.cn (D.L.); lvzhenyi@hrbmu.edu.cn (Z.L.); 2019020814@hrbmu.edu.cn (Y.Y.); 007881@hrbmu.edu.cn (L.W.); mabiao@hrbmu.edu.cn (B.M.); xuedongbo@hrbmu.edu.cn (D.X.); 2Department of General Surgery, The First Affiliated Hospital of Harbin Medical University, Harbin 150001, China

**Keywords:** chronic pancreatitis, trypsinogen activation, Rabep1, endocytosis pathway, pancreatic stellate cells activation

## Abstract

Background: The pathogenesis of chronic pancreatitis is still unclear. Trypsinogen activation is an active factor in acute pancreatitis that has not been studied in the occurrence of chronic pancreatitis. Methods: Immunofluorescence was used to detect the location and expression of trypsinogen in chronic pancreatitis and normal tissues. Microarray and single-cell RNA-seq (scRNA-seq) were used to screen core genes and pathways in pancreatic stellate cells (PSCs). Western blotting and immunofluorescence were used to verify trypsinogen expression in PSCs after silencing Rabep1. Immunofluorescence and flow cytometry were used to validate trypsinogen activation and PSC activation after intervening in the endocytosis pathway. Results: Endocytosed trypsinogen was found in PSCs in CP clinical samples. Bioinformatic analysis showed that Rabep1 is a core gene that regulates trypsinogen endocytosis through the endocytosis pathway, verified by Western blot and immunofluorescence. Immunofluorescence and flow cytometry analyses confirmed the activation of trypsinogen and PSCs through the endocytosis pathway in PSCs. Conclusion: This study discovered a new mechanism by which trypsinogen affects the activation of PSCs and the occurrence and development of CP. Through communication between pancreatic acinar cells and PSCs, trypsinogen can be endocytosed by PSCs and activated by the Rabep1 gene.

## 1. Introduction

Chronic pancreatitis (CP) is a pathological fibroinflammatory syndrome of the pancreas. Due to long-term inflammatory stimulation, pancreatic tissue atrophies and is gradually replaced by fibrotic tissue, resulting in a progressive loss of endocrine and exocrine functions of the pancreas. Clinically, patients often present with recurrent or persistent abdominal pain, diabetes (endocrine insufficiency), and dyspepsia (exocrine insufficiency), some of which will eventually progress to pancreatic cancer [1].

The premature activation of intracellular trypsin is considered to be the main cause of pancreatitis. Under normal physiology, trypsinogen is only activated upon contact with the intestinal brush border and its enzyme enterokinase [2]. Under pathological conditions, lysosomally hydrolyzed cathepsin B (CTSB) in pancreatic acinar cells is transferred into subcellular compartments through secretory vesicles, and converts trypsinogen into active trypsin in advance [2]. Activated trypsin will be phagocytosed by acinar cells, leading to damage to acinar, duct, and/or mesenchymal cells and inducing inflammatory cell infiltration and release of inflammatory cytokines, causing pancreatitis [3].

The more important pathological feature of chronic pancreatitis is irreversible pancreatic tissue fibrosis, which is mainly caused by the persistent activation of pancreatic stellate cells (PSCs) [3,4]. In the activated state, PSCs will proliferate and migrate to damaged areas, where they synthesize ECM components to promote pancreatic tissue repair [4]. Studies have shown that tissue necrosis and inflammatory processes are prerequisites for the activation of PSCs and that at the same time, activated PaSCs will produce autocrine factors that perpetuate the activated phenotype [4,5]. Therefore, the activation of trypsinogen and the activation of PSC are mutually promoting. 

Recently, trypsin activation was found to occur not only in pancreatic acinar cells but also in macrophages. Macrophages can endocytose vesicles containing trypsinogen from damaged acinar cells and activate them in a CTSB-dependent manner, thereby aggravating the inflammatory response [6]. In our study, we found that PSCs in CP tissues have abundant activated trypsinogen in vivo, and we wondered whether PSCs also have endocytosis of trypsinogen.

Endocytosis is the main process used by eukaryotic cells to take up extracellular material [7]. We were interested in RAB GTPase binding effector protein 1 (RABEP1) because it is an essential rate-limiting component in endosome fusion [8], and we found that it was significantly overexpressed in PSCs in CP, suggesting the activation of PSC endocytosis in CP. In the process of early endosome formation, RABEP1 synergistically activates Rab5 by stably combining with Rabex-5 and forming a complex that in turn regulates the docking and fusion of endosomal membranes, the movement of endosomes, and intracellular signal transduction. That is, RABEP1 is the important regulator and molecular switch of Rab5 function [9,10], and the overexpression of RABEP1 triggers the accumulation of endocytic vesicles. We speculate that PSCs can endocytose trypsinogen released from acinar cells, which involves endosome formation. The endocytosed trypsinogen can be activated in PSCs, aggravating pancreatic injury and inflammatory response and at the same time promoting PSC self-activation and aggravating CP fibrosis.

## 2. Methods and Materials

### 2.1. Clinical Samples

Tissues of 6 normal and 6 CP patients were obtained from the First Affiliated Hospital of Harbin Medical University.

Informed consent was obtained from patients, and the study was approved by the review committee of the First Affiliated Hospital of Harbin Medical University.

### 2.2. Reagents and Chemicals

#### Nafamostate and Bafilomycin-A1

Nafamostate (FUT-175, 81525-10-2, MCE, China) is a nonpeptide tetraguanidine benzoate compound that is a broad-spectrum serine protease inhibitor [11]. It has been clinically used in the treatment of pancreatitis because of its ability to inhibit trypsinogen activation [12,13].

Bafilomycin-A1 (88899-55-2, MCE, China), a macrolide antibiotic with a 16-membered lactone ring, is a specific inhibitor of the vacuolar H+-ATPase enzyme, and it functions to acidify the cell’s interior and transport protons across the plasma membrane. It can be used to inhibit the fusion of late endosomes and lysosomes [14]. The re-agent was also used as an autophagy inhibitor to inhibit the fusion of autophagosomes and lysosomes [15,16].

TLC (sodium taurocholate hydrate) (345909-26-4, MCE, China) is a cytotoxic substance, and its induction of pancreatic damage is dose-dependent [17].

Bovine trypsinogen was purchased from Yuanye Biotechnology Company (S27969, Yuanye, Shanghai, China). According to Matthias Sendler, exogenous trypsinogen can be used with a positive control group to exclude other interference factors of acinar cells [6].

### 2.3. Cell Culture and Treatment

AR42J: The rat pancreatic exocrine cell line AR42J was purchased from the American Type Culture Collection (Manassas, VA, USA) and cultured in Roswell Park Memorial Institute (RPMI) 1640 medium (Gibco, Grand Island, NY, USA) supplemented with 10% fetal bovine serum (ScienCell, San Diego, CA, USA), 100 U/mL penicillin, and 100 mg/mL streptomycin (Invitrogen, Carlsbad, CA, USA) at 37 °C in a 5% CO_2_ humidified incubator.

PSCs: Rat primary pancreatic stellate cells (PSCs) were purchased from PUHE Biotechnology Company LTD (Wuxi, China) and were cultured with special culture medium (CTCC-102MCM) for rat primary PSCs in 37 °C with 5% CO_2_.

There are five cell-cultured groups:

Group A: Control group AR42J were directly co-cultured with PSCs in Transwell Chambers for 6 h;

Group B: Experimental group AR42J were treated with 200 uM TLC for 40 min, then replaced with fresh medium and co-cultured with PSCs in Transwell Chambers for 6 h (upper compartment is PSCs, lower compartment is AR42J).

Group C: Positive control group PSCs were directedly treated with 10 ug/mL exogenous bovine trypsinogen (S27969, Yuanye, Shanghai, China) [6,18] for 6 h.

Group D: Trypsin suppression group AR42J (TLC-treated) were co-cultured with PSCs [Nafamostate(50 mmol/L) treated)] for 6 h.

Group E: Endocytosis inhibited group AR42J (TLC-treated) were co-cultured with PSCs [bafilomycin-A1 (100 nmol/L) treated] for 6 h.

For group B, part of the co-cultured PSCs were taken out. Removed the culture medium and the lysate was added at the ratio of 1 mL TRIzol per 10 cm^2^ cells. After full dissolution, the PSCs were put into the cryopreserved tube and transported to OE Biotech Co., Ltd. (Shanghai, China) for microarray detection with dry ice.

For subsequent flow cytometry, the cells were digested with 0.25% trypsin at room temperature for 2 min, terminated with medium, and centrifuged at 1000 rpm for 5 min. The supernatant was removed and washed with 1×PBS twice. After centrifugation at 1000 rpm for 5 min each time, the supernatant was discarded, and 100 uL PBS was added for flow cytometry.

Cells for subsequent confocal detection were washed twice with 1×PBS for backup.

### 2.4. Immunofluorescence

Tissues or cells were fixed with 4% paraformaldehyde, stabilized in 0.2% Triton X-100 for 10 min until cell membrane rupture, washed with PBS 3 times, and immersed in 2% BSA for 30 min to inhibit nonspecific antigen binding sites. The tissues were then incubated with anti-trypsinogen (1:500, Abcam, Cambridge, UK, ab166898) antibodies and anti-alpha smooth muscle actin (1:500, Abcam, Cambridge, UK, ab124964) overnight at 4 °C. The cells were then incubated with anti-trypsinogen (1:500, Abcam, Cambridge, UK, ab166898) antibodies or anti-alpha smooth muscle actin (1:500, Abcam, Cambridge, UK, ab124964) overnight at 4 °C. After washing, the tissues or cells were incubated with secondary antibody (Invitrogen) for 60 min, and the nuclei were stained with DAPI (Invitrogen) for 2 min. Then, the cells were washed with PBS and shielded from light before observation with a fluorescence microscope.

### 2.5. Datasets and Bioinformatics Analysis

Three pairs of primary PSCs cocultured with AR42J cells (with and without TLC intervention) were detected by microarray. Differentially expressed genes were identified between the two groups by the limma package (3.46.0) of R software (4.0.2). The threshold value (|log2FC| > 1 and *p* < 0.05) of genes indicated significant differences in expression. Genetic Ontology (GO) and Kyoto Encyclopaedia of Genes and Genomes (KEGG) analyses of differentially expressed genes were performed by the ClusterProfiler package (3.18.1) of R software (4.0.2), and *p* < 0.05 was considered significant.

Chronic pancreatitis scRNA-seq data GSE165045 including 5 CP patients (BL1–5) and 3 organ donors (BL6–8). We downloaded part of the CITE-seq (BL1–BL8: GSM5025052–GSM5025059) from the dataset, including 33,537 genes and 19,848 cells. The Seurat package SCTransform (0.3.3) function was used to preprocess and reduce the batch effect to integrate the single-cell transcriptome datasets. The most changed 3000 genes were chosen by SelectIntegrationFeatures, and the FindCluster package was used for cell cluster analysis with the resolution set to 0.15.

### 2.6. GeneMANIA

GeneMANIA (http://www.genemania.org (accessed on 20 June 2022)) is a user-friendly website that provides information on protein and genetic interactions, pathways, coexpression, colocalization, and protein domain similarity of submitted genes [19]. GeneMANIA was used for the analysis of Rabep1 co-expressed genes in human study from many public databases (such as GEO, BioGRID, and IRefIndex) and tissue or cell samples. Rabep1 was directly input into the search box of GeneMANIA database to obtain the network diagram of co-expressed genes.

### 2.7. Cell Transfection

Rabep1 in PSCs was inhibited by small interfering RNA (siRNA) and siRNA synthesized by Generalbiol Co. (Anhui, China). The siRNA sequences are listed in the Table 1. According to the manufacturer’s instructions, Lipofectamine 2000 Transfection Reagent (Gibco 11668019, Carlsbad, Calif, USA) and Opti-MEM (Gibco 31985070, Carlsbad, Calif, USA) were used to transfect siRNA at a final concentration of 100 nM. After transfection for 4 h, we replaced with conventional culture medium. Proteins were extracted 72 h after the medium replacement.

### 2.8. Western Blot

Multiple changes in trypsinogen expression in PSCs were detected by Western blot. Protein lysates were extracted from PSC cells using pyrolysis buffer (Beyotime) and quantified and isolated on 10% SDS–PAGE gels (Invitrogen). Rat primary antibodies against Rabep1 (1:1000, Abcam, Cambridge, UK), trypsinogen (1:1000, Abcam, Cambridge, UK) and β-actin (1:10,000, Abcam, Cambridge, UK) were used to imprint the protein onto a PVDF membrane (Millipore). HRP goat anti-rat IgG (1:20,000, Boster Biotechnology, Pleasanton, CA, USA) was incubated for 45 min as the secondary antibody. Strip visualization was performed by an enhanced chemiluminescence (ECL) system.

### 2.9. Flow Cytometry

PSC cells were stained with rhodamine 110, bis-(CBZ-L-isoleucyl-L-prolyl-L-arginine amide) dihydrochloride (BZiPAR, 5 mM; Invitrogen, Waltham, MA, USA) and anti-alpha smooth muscle actin (1:100, Abcam, Cambridge, UK). The expression levels of BZiPAR and trypsinogen were detected by a CytoFLEX flow cytometer (Beckman Coulter, Brea, CA, USA). Kaluza Flow Analysis software (Beckman Coulter, Brea, CA, USA) was used for data analysis.

### 2.10. Statistical Methods

All experimental data were analysed using GraphPad Prism 8.0 software. All data are presented as the mean ± standard deviation ($\bar{x}\pm sd$). Differences between the groups were analysed using one-way analysis of variance (ANOVA). A *p* value of less than 0.05 indicated that the difference was statistically significant.

## 3. Results

### 3.1. Endocytosis of Trypsinogen in PSCs Occurred in CP Samples

Trypsinogen has rarely been investigated in previous studies on CP. Immunofluorescence experiments have been performed on the pancreatic tissues of CP patients, and trypsinogen and PSC (α-SMA) were simultaneously labelled, with significant colocalization found between them. As shown in Figure 1, trypsinogen (red-labelled) was present in the PSCs (green-labelled)and was significantly overexpressed in the CP tissue compared with the control sample. The results suggested that endocytosis of trypsinogen occurred in CP in PSCs. Then, we further studied the effects and pathways of trypsinogen endocytosis in PSCs.

### 3.2. Bioinformatics Screening and Analysis in Microarray

We cocultured AR42J with PSCs and treated AR42J cells with TLC (sodium taurocholate hydrate) in advance as the experimental group. Gene expression profiles were detected and analysed. As shown in Figure 2, we screened 27 significantly differentially expressed genes (|log2FC| > 1 and *p* < 0.05). Rabep1 was significantly upregulated among them (log2FC = 2.27, *p* = 0.01). Rabep2 and Rab5a were coexpressed with this gene in the Genemania database (Figure 2B, http://genemania.org/ (accessed on 20 June 2022)). The results of the top 10 GO enriched terms are shown in Figure 2C, with the GO-CC showing significant enrichment of recycling endosome components (*p* < 0.05). The results of the first 30 KEGG enriched pathways are shown in Figure 2D, with the endocytosis pathway showing significant enrichment (*p* < 0.05). Interestingly, Rabep1 is included in this pathway. Based on the results above, we hypothesized that PSCs endocytose trypsinogen from pancreatic acinar cells through the endocytosis pathway mediated by Rabep1.

### 3.3. Endocytosis Pathway Confirmed in scRNA-seq

Endocytosis in the PSC of chronic pancreatitis has never been studied before, so we reconfirmed the existence of this pathway with single-cell sequencing data. The scRNA-seq (GSE165045) was used to describe the diversity and heterogeneity of various cell subsets in chronic pancreatitis tissues. Uniform manifold approximation and projection (UMAP) is used for nonlinear dimensionality reduction, and the cells are clustered by the FindCluster function; finally, 27 clusters are obtained (Figure 3A). Cell populations were classified annotated by specific gene expressions that were inferred from various human single immune cell RNA-seq studies, in which Clusters 22, 26, and 27 are labeled as PSCs (marker: MGP, SPARC, APOD, Figure 3B).

We further analyzed the frequency and composition of immune populations within each group or individual subject by identified clusters (Figure 3E). BL1-5 belongs to the sample of patients with chronic pancreatitis, while BL6-8 belonged to the sample of the control group. Most of the cells detected in this dataset were immune cells, and the number of PSCs was small. Next, we examined specific gene characteristics of each cell subpopulation by identifying differentially expressed genes (DEG) in CP versus control. The main representative genes of PSCs are SPARC, TAGLN, MGP, and LUM (Figure 3C). DEGs were further identified for the three clusters of PSCs, and it was found that Rab10 and Rabep1 were significantly enriched in Cluster27 (Figure 3D). Therefore, we conducted KEGG functional enrichment analysis on all DEGs in Cluster27 and found that the endocytosis pathway was included in the top 20 pathways (Figure 3F).

These results confirmed the enrichment of Rab10 and Rabep1 in PSCs and the existence of endocytosis pathway from the level of single cell subsets.

### 3.4. Regulation of Trypsinogen Endocytosis by Rabep1 in PSC

First, we detected the protein expression of Rabep1 in rat primary PSCs (cocultured with AR42J). Western blot results showed significantly high Rabep1 expression after treatment with TLC and significantly lower expression after silencing. As expected, trypsinogen was significantly higher in PSCs treated with TLC than in the control group but recovered after silencing Rabep1 (Figure 4A,B). Meanwhile, an immunofluorescence assay of trypsinogen showed that trypsinogen was significantly overexpressed in PSCs treated with TLC, while the expression level was recovered and significantly decreased after silencing Rabep1 (Figure 4C,D). These results suggested that in CP, high Rabep1 expression can lead to the activation of the endocytosis pathway in PSCs and result in an increase in endocytic trypsinogen.

### 3.5. Detection of Trypsinogen Endocytosis and Activation in PSCs

To explore whether trypsinogen endocytosis can also be activated in PSCs and to determine the activation mechanism, we established several intervention groups (Method 2.3). We assessed the endocytosis of trypsinogen with the expression of trypsinogen in PSCs (Figure 5A) and the activation of trypsinogen with the expression of bis-(CBZ-L-isoleucyl-L-prolyl-L-arginine amide) dihydrochloride (BZiPAR) in PSCs (Figure 5B) [20,21].

Immunofluorescence assay of trypsinogen showed that the endocytosis of trypsinogen in PSC was significantly increased in the TLC group, exogenous trypsinogen group, Nafa + TLC group, and Baf + TLC group compared with the control group (Figure 5A, *p* < 0.001). Immunofluorescence of BZiPAR showed a significant increase in trypsin activation in each group compared with the control group as well, but trypsin activation levels were significantly down-regulated in Nafa + TLC and Baf + TLC groups compared with the TLC group (Figure 5B, *p* < 0.001). The same results were confirmed by flow cytometry assay of BZiPAR (Figure 5C, *p* < 0.001). These results suggest that PSCs can activate endocytosed trypsinogen, which requires endosomal transport and fusion with lysosomes.

### 3.6. Activated Trypsinogen Provides Another Way to Activate PSCs

Sho Endo et al. pointed out that autophagy is required to activate PSCs and promote the growth of pancreatic tumours in mice [22]. In autophagy, the substrate protein is degraded in lysosomes by the fusion of the lysosome membrane and autophagosomes. The autophagy process is similar to the degradation of endocytosed trypsinogen in lysosomes. Therefore, we proposed that activated trypsinogen would also activate PSCs.

We assessed the activation of PSCs by the expression of α-smooth muscle actin (α-SMA) [23,24]. The results of immunofluorescence and flow cytometry showed that α-SMA expression was significantly increased after treatment with TLC and trypsinogen compared with the control group (Figure 6A, *p* < 0.001). As expected, when we blocked the activation of trypsinogen with Nafa or Baf, the expression of α-SMA showed a significant decrease compared with TLC and trypsinogen (Figure 6A, *p* < 0.001). The same results were confirmed by a flow cytometry assay of α-SMA (Figure 6B, *p* < 0.001). These results suggest that activated trypsinogen provides another method of activating PSCs, and the specific mechanism should be further studied.

### 3.7. Discussion

CP is a chronic inflammatory and fibrotic disease of the pancreas. Both genetic and environmental factors can cause CP. Many studies have recognized that activated PSCs play a vital role in the progression of CP [5,25].

A recent article reported that trypsinogen activation in mice is not only essential for the occurrence and development of acute pancreatitis but also can lead to the occurrence and development of CP [26]. Professor Andrea Geisz created a mouse model of CP by mutation of cationic trypsinogen (T7 subtype) T7D23A. This remarkable model provides in vivo evidence that trypsinogen self-activation can drive the pathogenesis and development of CP [27].

Atsushi Masamune et al. found that trypsin can induce PSC proliferation and collagen production in rats by activating protease-activated receptor-2 [28]. It has been reported that macrophages recruited from acute pancreatitis can swallow and digest trypsin-containing vesicles. Lysosomal cathepsin B interferes with phagocytosed zymogen, resulting in the activation of zymogen and the aggravation of acute pancreatitis [6]. A similar phenomenon was also found in the preliminary experiments for this study. Trypsinogen endocytosis was found in the PSCs of pancreatic tissue through immunofluorescence. However, PSCs do not synthesize or secrete trypsinogen by themselves. We have known that trypsinogen is associated with the secretion of pancreatic acinar cells. How does the secreted trypsinogen enter the PSC? Does it affect the pancreatic fibrosis of CP?

To study the specific mechanism, we screened for the fact that Rabep1 was significantly upregulated and that recycling endosome components were significantly enriched. The endocytosis pathway [29,30] was significantly enriched and contained the Rabep1 gene. The endocytosis pathway mediates a variety of intracellular regulatory effects, such as nutrient uptake, receptor internalization and cell signalling. Many bacteria, viruses and toxins also enter cells through this pathway. After endocytosed vesicles enter the cell, Rab GTPases are rapidly recruited by effector proteins, which mediate the formation of early endosomes. During this process, Rabep1 interacts with Rab5, thus leading to an increase in the uptake of endocytic substances [31]. Yi Wang et al. also found that Rabep1, as a key effector molecule of Rab5, can affect the efficiency of cell endocytosis by regulating the early endosome fusion mediated by Rab5 [32]. Therefore, these results suggest that trypsinogen secreted by pancreatic acinar cells enters PSCs through the clathrin-mediated endocytosis pathway and rapidly forms early endosomes through Rabep1. To verify these results, we designed a follow-up cell-silencing experiment. Through Western blot, we observed that TLC treatment of AR42J cells increased trypsinogen levels in PSCs, while siRNA-Rabep1 transfection in PSCs reduced trypsinogen levels in PSCs. This result was confirmed in immunofluorescence experiments. These results fully verified that Rabep1 can regulate PSC endocytosis of trypsinogen.

Studies have confirmed that when pancreatitis occurs, stress stimulates autophagy in acinar cells. Then, the trypsinogen granules are wrapped by autophagosomes and enter the lysosome. At this stage, they are susceptible to the interference of lysosomal cathepsin B, which activates the transformation of trypsinogen to trypsin in advance. Moreover, during inflammation, the apical secretion of trypsinogen under physiological conditions is reduced. Trypsinogen granules are mainly secreted into intercellular substances through the basement membrane to recruit inflammatory mediators and other cascades that aggravate inflammatory reactions [33]. In subsequent experiments, we found that PSCs can also activate endocytosed trypsinogen through a similar pathway by immunofluorescence and flow cytometry experiments. Early experiments found that Rabep1 mediates trypsinogen to form early endosomes. We speculate that the early endosomes formed by trypsinogen undergo intracellular transport and finally reach and are activated in the lysosome to form trypsin. In this study, these two inhibitors (nafamostate and bafilomycin-A1) were also used to inhibit the activation and transportation of trypsinogen. We found that compared with the TLC stimulation group, significant differences in the expression of trypsinogen were not observed in the Nafa + TLC group, while trypsin was significantly reduced, and the same result was observed in the Baf + TLC group. The results of this part confirm that PSCs can activate endocytosed trypsinogen. The process of trypsinogen activation requires endosome transport and fusion with lysosomes.

Activated trypsinogen in turn activates PSCs and ultimately aggravates the progression of CP. Alpha-SMA has been recognized as a marker of PSCs, and high expression of α-SMA indicates the activation of PSCs and the aggravation of CP [23,34]. We found that in the TLC stimulation group, α-SMA was significantly highly expressed, which suggested that PSCs were activated. However, in the Nafa + TLC group and the Baf + TLC group, this activation was significantly inhibited, which was confirmed by flow cytometry. There are some limitations in this study. First, mechanistic verification experiments have not been performed using animal and clinical samples. Second, the mechanistic experiments were only confirmed by the intervention of inhibitors, and certain interfering factors may have been involved. In the future, we will conduct in-depth research on the activation mechanism of trypsinogen in PSCs through animal and clinical samples and adopt diverse intervention methods to ensure that the research more rigorous.

This study discovered a new mechanism by which trypsinogen affects the activation of PSCs and the occurrence and development of CP. Through communication between pancreatic acinar cells and PSCs, trypsinogen secreted by acinar cells enters PSCs. Endocytosed trypsinogen is activated after reaching the lysosome through the intracellular transport process mediated by the Rabep1 gene. Finally, activated trypsinogen leads to the activation of PSCs. The results of this study provide new insights into the pathogenesis of CP and new ideas for early interventions in the occurrence and development of CP.

## Figures and Tables

**Figure 1 biomolecules-12-01063-f001:**
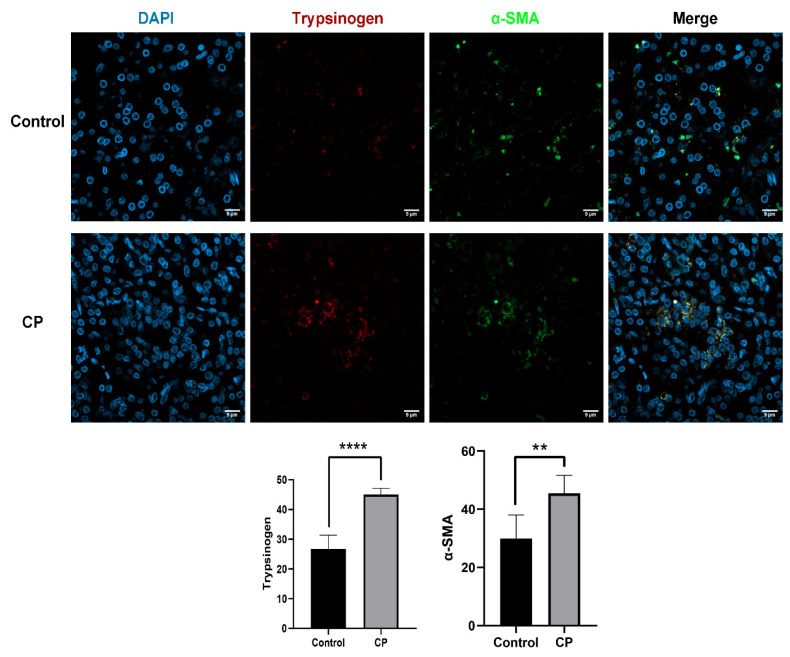
Endocytosis of trypsinogen in PSC occurred in CP. Immunofluorescence microscope was used to detect the expression and location of trypsinogen and pancreatic stellate cells (PSC) in the control group and chronic pancreatitis (CP) tissues. Blue indicates DAPI, red indicates trypsinogen, and green indicates PSC. Scale bars = 9 μm. ** *p* < 0.01, **** *p* < 0.0001.

**Figure 2 biomolecules-12-01063-f002:**
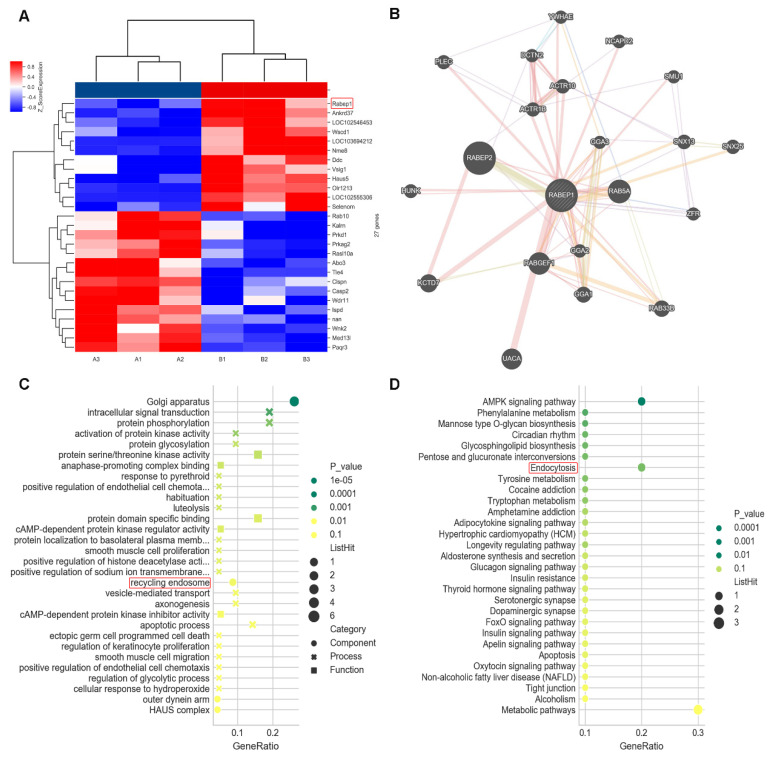
Rabep1 is highly expressed in CP and is related to the endocytosis pathway: (**A**) according to the Microarray data, Rabep1 showed higher expression level in CP group (B1−B3, treated with TLC) compared with control group (A1−A3). (**B**) Analysis of Rabep1 co-expressed genes through genemania database. (**C**) GO enrichment analysis found that Rabep1 is closely related to the recycling endosome. (**D**) KEGG enrichment analysis showed that Rabep1 was significantly enriched in Endocytosis pathway.

**Figure 3 biomolecules-12-01063-f003:**
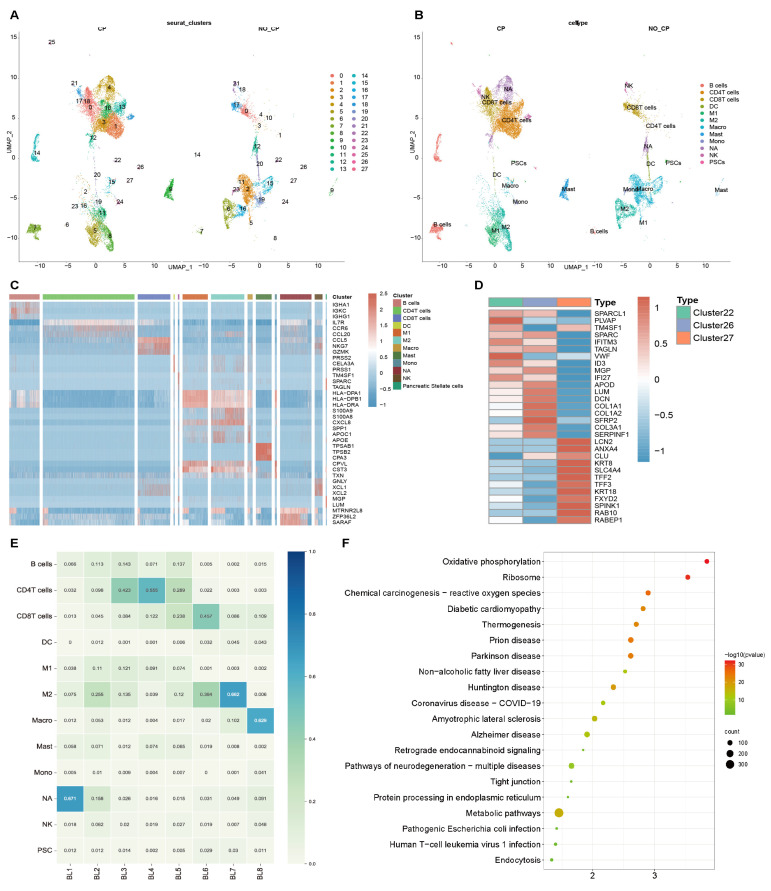
The endocytosis pathway is verified in PSCs with scRNA-seq. (**A**,**B**) UMAP plots of all 19848 pancreatic cells classified by Seurat clusters (**A**) and cell types (**B**). (**C**) Heatmap of significant marker gene expressions in all clusters. (**D**) Heatmap of significant marker gene expressions in PSCs, including clusters 22, 26, and 27. (**E**) The percentages of subsets of cells in each sample. (**F**) KEGG enrichment analysis showed that Endocytosis was in top20 pathways.

**Figure 4 biomolecules-12-01063-f004:**
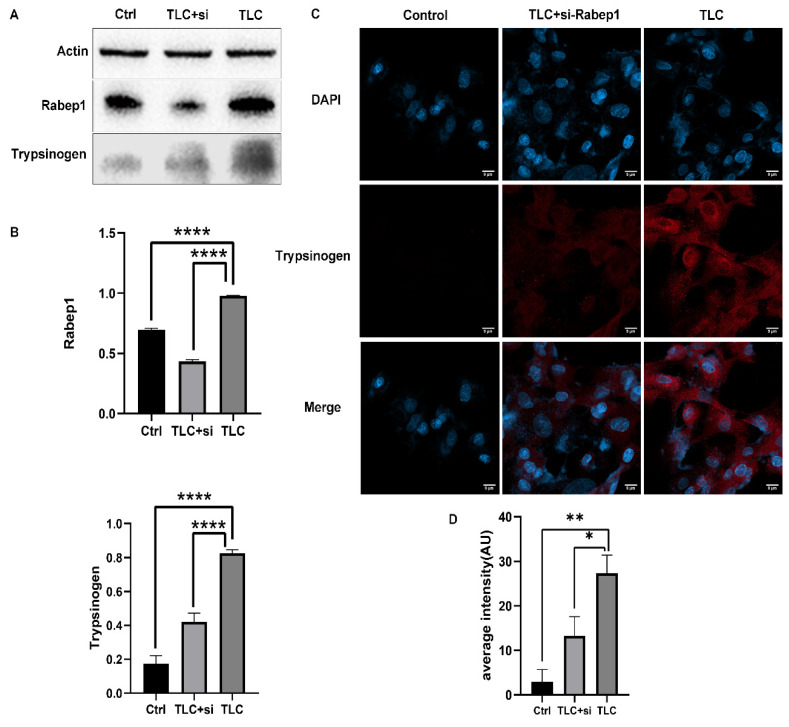
Rabep1 regulates the endocytosis of trypsinogen in PSCs. (**A**,**B**) Western blot analysis found that trypsinogen was highly expressed in TLC-treated PSC cells and that silencing Rabep1 inhibited it. (**C**,**D**) Trypsinogen expression was assessed by immunofluorescence in PSC treated with TLC or (and) silencing. Rabep1, Blue indicates DAPI, red indicates trypsinogen. Scale bars = 9 μm. * *p* < 0.05, ** *p* < 0.01, **** *p* < 0.0001.

**Figure 5 biomolecules-12-01063-f005:**
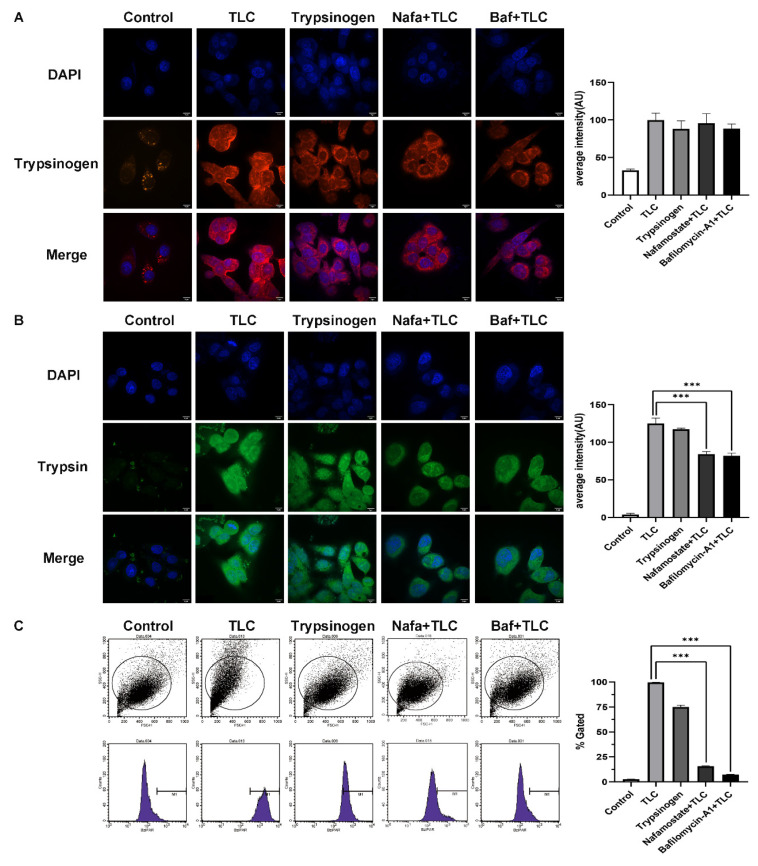
Trypsinogen is activated through the endocytosis pathway in PSC. (**A**) Trypsinogen expression level was assessed by immunofluorescence in PSC treated with TLC or trypsinogen or nafamostate or bafilomycin-A1. Blue indicates DAPI, red indicates trypsinogen. Scale bars = 9 μm. (**B**) Trypsinogen activation level was assessed by immunofluorescence in PSC treated with TLC or trypsinogen or nafamostate or bafilomycin-A1. Blue indicates DAPI, green indicates active trypsinogen. Scale bars = 9 μm. (**C**) Trypsinogen activation level was verified by flow cytometry experiment. *** *p* < 0.001.

**Figure 6 biomolecules-12-01063-f006:**
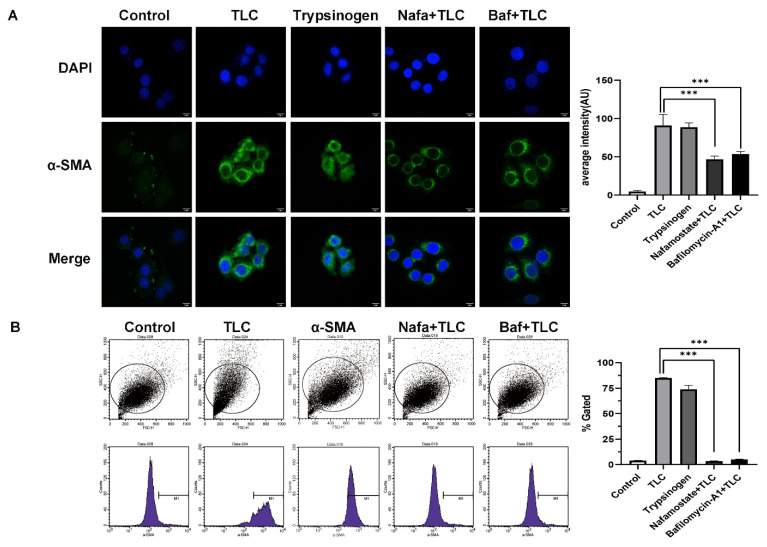
Activated trypsinogen provides another way to activate PSC. (**A**) α-SMA expression level, used to assess the activation degree of PSC, was assessed by immunofluorescence in PSC treated with TLC or trypsinogen or nafamostate or bafilomycin-A1. Blue indicates DAPI, green indicates active PSC. Scale bars = 9 μm. (**B**) α-SMA expression level was verified by flow cytometry experiment. *** *p* < 0.001.

**Table 1 biomolecules-12-01063-t001:** Sequence of siRNA-Rabep1 designed by Generalbiol Co.

PIN	Primer Name	Sequence (5′ to 3′)	Length	MW (g/mol)	Tm °C	GC %	Purification
A2276696	Rabep1 (Norway rat) siRNA-1962	CAGAAGAGCUGGUGAGGUUTT	21	6789.21	52.4	47.6	HPLC
A2276697		AACCUCACCAGCUCUUCUGTT	21	6525.99	52.4	47.6	HPLC

## Data Availability

Publicly available datasets were analyzed in this study. Data can be found at GEO dataset (https://www.ncbi.nlm.nih.gov/geo (accessed on 20 June 2022)).

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
