# Peer review of "The Rabep1-Mediated Endocytosis and Activation of Trypsinogen to Promote Pancreatic Stellate Cell Activation"

_biomolecules, 2022, doi:10.3390/biom12081063_

Round 1
Reviewer 1 Report
Chronic pancreatitis (CP) is a progressive inflammatory disease with increasing incidence, which severely decrease of the quality of life of patients due to the debilitating pain and exocrine pancreatic insufficiency resulting in malabsorption. The disease pathogenesis is a complex mechanism, however incompletely understood. The medical treatment of CP is limited to supportive therapy, including enzyme replacement and analgetics. It is well known, that genetic mutations affected to trypsinogen lead to intraacinar or intrapancreatic activation of the digestive enzyme cascade resulting in the autodigestion of the tissue.It is also known, that activation of quiescent pancreatic stellate cells (PSCs) into an activated, alpha-Smooth Muscle Actin (a-SMA) positive myofibroblast-like phenotype will lead to excessive production and deposition of extracellular matrix proteins, eventually leading to fibrosis.
Based on these facts, I think the this research topic is absolutely actual and relevant and the results from this study contribute to understand the role of trypsinogen activation in the activation of pancreatic stellate cells.
comments:
I think the description of all parts of the methods is incomplete. For example how did the pancreatic stellate cells isolated? I think it is a relevant information Becuase of the isolation of PSC is very difficult because of the activation of the cells during the isolation process.
Figure 1: The size of immunfluorescent pictures is too small, hence poorly visible. It seems like the alpha SMA staining is diffuse and no specific. The authors should change the picture of IF staining to an other ones with higher resolution. The quantification of the IF staining is missing, but it would be useful.
Fig 2. What does each groups (A1-3, B1-3) mean? The Figure legend of figure 2 should be contain more details because in this from the figure is incomprehensible.
Figure3. Which samples did you used for the flow cytometry? Where are the CP samples from? From surgical resection?
Figure4. It should be useful if the authors quantify the result of Western blott.
Figure 5. I think the desciption of the experiments is incomplete so a little bit hard to understand. What does BZiPAR labelled?
Important informations are missing from the text. How long were the PSC incubated with Nafa and Baf? Which concentrations of Nafa and Baf did the authors used?
Author Response
Reviewer 1:
The figure1, 2 can not show here, so we have uploaded a word version as well.
- I think the description of all parts of the methods is incomplete. For example how did the pancreatic stellate cells isolated? I think it is a relevant information Becuase of the isolation of PSC is very difficult because of the activation of the cells during the isolation process.
Reply: Thanks for your comments, we have modified and improved the Method. “Clinical samples, Reagents and chemicals, Cell culture and treatment” were added to the Method part. Actually, Rat primary pancreatic stellate cells (PSCs) were purchased from PUHE Biotechnology Company LTD(Wuxi, China).
Here is the detailed process of PSCs extraction method: After removed from the neck to death, the rats were immersed in alcohol for disinfection. The abdominal cavity was cut open along the midline with scissors, the pancreas was cut out along the duodenum and put into PBS containing double antibiotics. At the same time, the excess fat and connective tissue were cut out. The pancreas was cut into 1 mm3 pieces and put into EP tube containing 0.05% collagenase. The EP tube was placed in an incubator at 37℃ for digestion for 30 min. The digested tissues were blown with a pipettor, left for 1 min, then supernatant was absorbed, centrifuged at 1200 RPM /min for 3 min, supernatant was discarded after centrifugation, and inoculated in a dish with complete medium. At last immunofluorescence detection of Desmin was performed. (figure 1)
Figure 1. Immunofluorescence detection of Desmin for primary PSCs.
As you said, “the isolation of PSC is very difficult because of the activation of the cells”. The instructions also indicated that PSC cells started the activation process 24 hours after inoculation, and most cells were activated on day 6 of culture. After subculture, the cells were in a highly activated state. Therefore, we immediately carried out co-culture experiment after receiving the primary PSC cells.
- Figure 1: The size of immunfluorescent pictures is too small, hence poorly visible. It seems like the alpha SMA staining is diffuse and no specific. The authors should change the picture of IF staining to an other ones with higher resolution. The quantification of the IF staining is missing, but it would be useful.
Reply: Your observation of the results is very careful. We re-analyzed the immunofluorescence image. Here we uploaded the original high resolution images and added quantitative analysis to the results (figure 2). As you said, “the alpha SMA staining is diffuse and no specific”. We think that this may be due to the limited number of PSC cells in CP tissue. The obvious co-localization in the composite image indicated that this is indeed specific staining. However, we did not find cells with typical PSC morphology, so further studies in this area need to be conducted in the future.
Figure 2. Endocytosis of trypsinogen in PSC was occurred in CP. Immunofluorescence microscope was used to detect the expression and location of trypsinogen and pancreatic stellate cells (PSC) in the control group and chronic pancreatitis (CP) tissues. Blue indicates DAPI, red indicates trypsinogen, and green indicates PSC. Scale bars = 9μm.
- Fig 2. What does each groups (A1-3, B1-3) mean? The Figure legend of figure 2 should be contain more details because in this from the figure is incomprehensible.
Reply: We are sorry that the legend information of Figure 2 is not complete, we have modified it. A1-3 represents co-cultured PSCs in NC group, and B1-3 represents co-cultured PSCs (treated with TLC) in CP group.
- Which samples did you used for the flow cytometry? Where are the CP samples from? From surgical resection?
Reply: We are sorry that we didn't make clear the data source of the Figure 3 result. The single-cell sequencing data were from GSE165045 dataset included in GEO database, Their sample sources contained “organ donors (n=3) and CP patients (n=4) who underwent total pancreatectomy”[1].
- It should be useful if the authors quantify the result of Western blot.
Reply: Thank you for your suggestions on this part of the results. We have quantified the results in the modified manuscript and the quantified results are consistent with the description before.
- Figure 5. I think the desciption of the experiments is incomplete so a little bit hard to understand. What does BZiPAR labelled?
Reply: We are sorry for giving you a bad reading experience, we have revised this part of the results description, and have marked it in the modified manuscript. BZiPAR was labeled as activated trypsin, as used in the study of Mark W. Sherwood, the co-localization of a liquid tracer with lysed BZiPAR can indicate intracellular activation of trypsinogen [2, 3]. Results in this part showed not only endocytosis of trypsinogen (Fig 5A) but also activation of trypsinogen (Fig 5B,C).
Here is the modified paragraph:
To explore whether trypsinogen endocytosis can also be activated in PSCs and to determine the activation mechanism, we established several intervention groups (Method 2.3). We assessed the endocytosis of trypsinogen with the expression of trypsinogen in PSCs (Fig 5A) and the activation of trypsinogen with the expression of bis-(CBZ-L-isoleucyl-L-prolyl-L-arginine amide) dihydrochloride (BZiPAR) in PSCs (Fig 5B) [2, 3].
Immunofluorescence assay of trypsinogen showed that the endocytosis of trypsinogen in PSC was significantly increased in the TLC group, exogenous trypsinogen group, Nafa+TLC group and Baf+TLC group compared with the control group (Figure 5A, p<0.001). Immunofluorescence of BZiPAR showed a significant increase in trypsin activation in each group compared with the control group as well, but trypsin activation levels were significantly down-regulated in Nafa+TLC and Baf+TLC groups compared with the TLC group (Fig 5B, p<0.001). The same results were confirmed by flow cytometry assay of BZiPAR (Fig 5C, p<0.001). These results suggest that PSCs can activate endocytosed trypsinogen, which requires endosomal transport and fusion with lysosomes.
- Important informations are missing from the text. How long were the PSC incubated with Nafa and Baf? Which concentrations of Nafa and Baf did the authors used?
Reply: We are very sorry for the lack of a lot of important information in our method, which has brought a lot of inconvenience to your review. We have added the details of drug treatment in the modified manuscript.
In the trypsin inhibition group, AR42J cells (treated with TLC) were co-cultured with PSCs [treated with Nafamostate(50mmol/L)] for 6h.
In the endocytosis inhibition group, AR42J cells (treated with TLC) were co-cultured with PSCs [treated with Bafilomycin-A1 (100nmol/L)] for 6h.
Reference:
- Lee, B., Namkoong, H., Yang, Y., Huang, H., Heller, D., Szot, G. L., Davis, M. M., Husain, S. Z., Pandol, S. J., Bellin, M. D., & Habtezion, A. (2021). Single-cell sequencing unveils distinct immune microenvironments with CCR6-CCL20 crosstalk in human chronic pancreatitis. Gut, gutjnl-2021-324546. Advance online publication. https://doi.org/10.1136/gutjnl-2021-324546
- Sherwood, M. W., Prior, I. A., Voronina, S. G., Barrow, S. L., Woodsmith, J. D., Gerasimenko, O. V., Petersen, O. H., & Tepikin, A. V. (2007). Activation of trypsinogen in large endocytic vacuoles of pancreatic acinar cells. Proceedings of the National Academy of Sciences of the United States of America, 104(13), 5674–5679. https://doi.org/10.1073/pnas.0700951104
- Ma, B., Wu, L., Lu, M., Gao, B., Qiao, X., Sun, B., Xue, D., & Zhang, W. (2013). Differentially expressed kinase genes associated with trypsinogen activation in rat pancreatic acinar cells treated with taurolithocholic acid 3-sulfate. Molecular medicine reports, 7(5), 1591–1596. https://doi.org/10.3892/mmr.2013.1355

Reviewer 2 Report
Premature intracellular activation of trypsinogen to trypsin is an important pathophysiological process in acute pancreatitis, but its involvement in chronic pancreatitis is not clear. In this study, W. Yao et al. investigated trypsinogen endocytosis and activation in human CP and in rat pancreatic stellate cells (PSCs). The authors utilized human CP samples and online single cell-sequencing data sets, and in vitro acinar and PSC co-culture and treatment with inhibitors. They demonstrated association of Rabep1 and the endocytosis pathway with CP, and Rabep1 mediated trypsinogen endocytosis in PSCs by Rabep1 knockdown. They conclude that trypsinogen released from pancreatic acinar cells can be endocytosed and activated by PSCs through Rabep1, which further activated PSCs contributing to CP. Overall, this study explored the novel idea of trypsinogen endocytosis and activation in CP. However, there are several major issues on the experimental design, in addition to multiple inconsistency throughout the manuscript.
Major comments:
1. Results 3.1 and Fig. 1: how many human control (normal) and CP samples were used and the source for immunofluorescence (IF)? What were the staining patterns of trypsinogen in control and CP, and respective quantification?
2. Results 3.3 and Fig. 3F: the text description on line 202-203 is not consistent with Fig. 3F legend.
3. Results 3.5 and 3.6 and Fig. 5 and 6: What was the rationale for trypsinogen treatment and the source of trypsinogen? The text description on line 230-233 for Fig. 5 is confusing, particularly on statistics.
Other comments:
Abstract:
1. Is “Microarray” equal to “Chip-seq” on line 156?
2. Conclusion should be specified as “…trypsinogen can be endocytosed by PSCs…” on line 28.
Introduction:
1. CP is a major risk factor of pancreatic cancer, but not all CP “will eventually progress to pancreatic cancer” as stated on line 38.
2. “In our study, we found that PSCs in CP tissues have abundant activated trypsin in vivo…” on line 58-59, in fact only trypsinogen IF was shown in Fig 1.
3. A grammar error, “that internalization of PSCs in CP Activation of phagocytosis” on line 64.
4. “We speculate that PSCs can induce endocytosis of trypsinogen by PSCs by regulating early endosome formation. Further trypsinogen was induced…” on line 70-71, this raises two questions, 1) how can “PSCs induce… by PSCs”? 2) do PSCs express or produce trypsinogen or only endocytose trypsinogen that is produced and released by acinar cells?
Materials and Methods:
1. In 2.1, Please include other key reagents, such as TLC and trypsinogen.
2. In 2.2, “Tissue or cells” or “cells” are not consistently used.
3. In 2.3, Please provide description of the cell lines, co-culture and treatment, and microarray or Chip-seq. Several references were missing as (), which were also seen throughout the text.
4. In 2.4, Please specify the application of GeneMANIA for the study.
5. In 2.5, Were rat PSCs used and the source? The description is confusing, “After transfection for 4 h, proteins were extracted 72 h after transfection” on line 119-120.
6. In 2.8, “standard deviation” should be abbreviated as “sd” or “SD”.
Results:
1. In Fig. 3 legend: Please define BL1-8 as human samples, CP or control. Were these 8 human samples different from 12 human samples mentioned in Methods?
Discussion:
1. “We speculate that trypsinogen is associated with the secretion of pancreatic acinar cells” on line 283-284. This should be a fact, not a speculation.
2. “Nafamostate is a serine protease inhibitor… Bafilomycin-A1 is a vacuolar H+-ATPase inhibitor…” on line 316-319. This is unnecessary repeat of the description in 2.1. Materials.
Overall comments:
1. Abbreviations: should be spelled out completely on initial appearance in text, for example, RAB GTPase Binding Effector Protein 1 (Rabep1).
2. Figure labels: Should be consistent, either as Fig. or fig.
3. “Overexpression of trypsinogen”, and “rescued” are not appropriate to the content.
4. Missing ref as ().
5. In Funding: “This research received no external funding” or “This research was funded by NAME OF FUNDER, grant number XXX” and “The APC was funded by XXX”. Check carefully that the details given are accurate and use the standard spelling of funding agency names at https://search.crossref.org/funding. Any errors may affect your future funding” on line 351-354, is unnecessary.
Author Response
Reviewer 2:
Major comments:
- Results 3.1 and Fig. 1: how many human control (normal) and CP samples were used and the source for immunofluorescence (IF)? What were the staining patterns of trypsinogen in control and CP, and respective quantification?
Reply: We are sorry for the lack of these key information in the manuscript. We have modified and improved the methods and results of the manuscript. We used tissues of 6 normal and 6 CP patients from the First Affiliated Hospital of Harbin Medical University.
As shown in immunofluorescence of cells below, the staining pattern of Trypsinogen belongs to cytoplasmic fine spots, and α-SMA belongs to cytoskeletal staining pattern. Meanwhile, quantitative analysis of the figure 1 was conducted, and the results were consistent with the original description.
- Results 3.3 and Fig. 3F: the text description on line 202-203 is not consistent with Fig. 3F legend.
Reply: Thanks for your careful observation, we have modified the legend of figure 3F.
Here is the modified legend: KEGG enrichment analysis showed that Endocytosis was in top20 pathways.
- Results 3.5 and 3.6 and Fig. 5 and 6: What was the rationale for trypsinogen treatment and the source of trypsinogen? The text description on line 230-233 for Fig. 5 is confusing, particularly on statistics.
Reply: We are sorry for bringing you bad reading experience. We have revised this part of the result description and marked it in the modified manuscript. The bovine trypsinogen was purchased from Yuanye biotechnology Company (S27969, Yuanye, China). According to the study of Matthias Sendler, exogenous trypsinogen can be used to the positive control group, to exclude other interference factors of acinar cells [1]. BZiPAR was labeled as activated trypsin, as used in the study of Mark W. Sherwood, the co-localization of a liquid tracer with lysed BZiPAR can indicate intracellular activation of trypsinogen [2, 3].
Here is the modified paragraph:
To explore whether trypsinogen endocytosis can also be activated in PSCs and to determine the activation mechanism, we established several intervention groups (Method 2.3). We assessed the endocytosis of trypsinogen with the expression of trypsinogen in PSCs (Fig 5A) and the activation of trypsinogen with the expression of bis-(CBZ-L-isoleucyl-L-prolyl-L-arginine amide) dihydrochloride (BZiPAR) in PSCs (Fig 5B) [2, 3].
Immunofluorescence assay of trypsinogen showed that the endocytosis of trypsinogen in PSC was significantly increased in the TLC group, exogenous trypsinogen group, Nafa+TLC group and Baf+TLC group compared with the control group (Fig 5A, p<0.001). Immunofluorescence of BZiPAR showed a significant increase in trypsin activation in each group compared with the control group as well, but trypsin activation levels were significantly down-regulated in Nafa+TLC and Baf+TLC groups compared with the TLC group (Fig 5B, p<0.001). The same results were confirmed by flow cytometry assay of BZiPAR (Fig 5C, p<0.001). These results suggest that PSCs can activate endocytosed trypsinogen, which requires endosomal transport and fusion with lysosomes.
Other comments:
Abstract:
- Is “Microarray” equal to “Chip-seq” on line 156?
Reply: We are sorry for the inconsistent description. We have unified modified it into “Microarray”.
- Conclusion should be specified as “…trypsinogen can be endocytosed by PSCs…” on line 28.
Reply: Thank you for your careful and kind review, which is admirable. We have unified modified it into “…trypsinogen can be endocytosed by PSCs…”.
Introduction:
- CP is a major risk factor of pancreatic cancer, but not all CP “will eventually progress to pancreatic cancer” as stated on line 38.
Reply: We apologize for the imprecise description. We have modified to “some of which will eventually progress to pancreatic cancer”.
- “In our study, we found that PSCs in CP tissues have abundant activated trypsin in vivo…” on line 58-59, in fact only trypsinogen IF was shown in Fig 1.
Reply: We apologize for the imprecise description. We have modified to “In our study, we found that PSCs in CP tissues have abundant activated trypsinogen in vivo…”.
- A grammar error, “that internalization of PSCs in CP Activation of phagocytosis” on line 64.
Reply: We apologize for the grammar error. We have modified to “suggesting that activation of PSC endocytosis in CP”.
- “We speculate that PSCs can induce endocytosis of trypsinogen by PSCs by regulating early endosome formation. Further trypsinogen was induced…” on line 70-71, this raises two questions, 1) how can “PSCs induce… by PSCs”? 2) do PSCs express or produce trypsinogen or only endocytose trypsinogen that is produced and released by acinar cells?
Reply: We are sorry for the inconsistent description. According to our results, 1) PSCs cannot induce PSCs; 2) PSCs endocytosis only trypsinogen produced by acinar cells.
We have modified to “We speculate that PSCs can induce endocytosis of trypsinogen from acinar cells by regulating early endosome formation. Further trypsinogen was induced…”.
Materials and Methods:
- In 2.1, Please include other key reagents, such as TLC and trypsinogen.
Reply: We are sorry for the lake of other key reagents. We have modified and improved the experimental methods, including “2.2 Reagents and chemicals” and “2.3 Cell culture and treatment”.
TLC (sodium taurocholate hydrate) (345909-26-4, MCE, China) as a cytotoxic substance, its induction of pancreatic damage is dose-dependent, and can be used to make acute pancreatitis cell and animal models by controlling drug concentration [4].
Bovine trypsinogen was purchased from Yuanye biotechnology Company (S27969, Yuanye, China). According to the study of Matthias Sendler, exogenous trypsinogen can be used to the positive control group, to exclude other interference factors of acinar cells [5].
- In 2.2, “Tissue or cells” or “cells” are not consistently used.
Reply: Thank you for your comments. We have made modifications in the method.
Here is the modified paragraph:
Tissues or cells were fixed with 4% paraformaldehyde, stabilized in 0.2% Triton X-100 for 10 min until cell membrane rupture, washed with PBS 3 times, and immersed in 2% BSA for 30 min to inhibit nonspecific antigen binding sites. The tissues were then incubated with anti-trypsinogen (1:500, Abcam, ab166898) antibodies and anti-alpha smooth muscle actin (1:500, Abcam, ab124964) overnight at 4 °C. The cells were then incubated with anti-trypsinogen (1:500, Abcam, ab166898) antibodies or anti-alpha smooth muscle actin (1:500, Abcam, ab124964) overnight at 4 °C. After washing, the tissues or cells were incubated with secondary antibody (Invitrogen) for 60 min, and the nuclei were stained with DAPI (Invitrogen) for 2 min. Then, the cells were washed with PBS and shielded from light before observation with a fluorescence microscope.
- In 2.3, Please provide description of the cell lines, co-culture and treatment, and microarray or Chip-seq. Several references were missing as (), which were also seen throughout the text.
Reply: Thank you for your comments. We have made modifications in the method.
Here is the modified paragraph:
2.3. Cell culture and treatment
AR42J:The rat pancreatic exocrine cell line AR42J was purchased from the American Type Culture Collection (Manassas, VA, USA) and cultured in Roswell Park Memorial Institute (RPMI) 1640 medium (Gibco, Grand Island, NY, USA) supplemented with 10% fetal bovine serum (ScienCell, San Diego, CA, USA), 100 U/ml penicillin and 100 mg/ml streptomycin (Invitrogen, Carlsbad, CA, USA) at 37 °C in a 5% CO2 humidified incubator.
PSCs:Rat primary pancreatic stellate cells (PSCs) were purchased from PUHE Biotechnology Company LTD(Wuxi, China) and were cultured with special culture medium(CTCC-102MCM) for rat primary PSCs in 37°C with 5% CO2.
There are five cell cultured groups:
Group A: Control group, AR42J were directly co-cultured with PSCs in Transwell Chambers for 6h;
Group B: Experimental group, AR42J were treated with 200uM TLC for 40min, then replaced with fresh medium, and co-cultured with PSCs in Transwell Chambers for 6h (upper compartment is PSCs, lower compartment is AR42J).
Group C: Positive control group, PSCs were directedly treated with 10ug/mL exogenous bovine trypsinogen (S27969, Yuanye, China) [5, 6] for 6h.
Group D: Trypsin suppression group, AR42J (TLC-treated) were co-cultured with PSCs [Nafamostate(50mmol/L) treated)] for 6h.
Group E: Endocytosis inhibited group, AR42J (TLC-treated) were co-cultured with PSCs [Bafilomycin-A1 (100nmol/L) treated] for 6h.
For the group B, part of the co-cultured PSCs were taken out. Removed the culture medium and the lysate was added at the ratio of 1mL TRIzol per 10cm2 cells. After full dissolution, the PSCs were put into the cryopreserved tube and transported to OE Biotech Co., Ltd. (Shanghai, China) for Microarry detection with dry ice.
For subsequent flow cytometry, the cells were digested with 0.25% trypsin at room temperature for 2min and terminated with medium, centrifuged at 1000rpm for 5min. The supernatant was removed and washed with 1*PBS twice. After centrifugation at 1000rpm for 5min each time, the supernatant was discarded and 100ul PBS was added for flow cytometry.
Cells for subsequent confocal detection were washed twice with 1*PBS for backup.
- In 2.4, Please specify the application of GeneMANIA for the study.
Reply: I am sorry that we did not specify the application of GeneMANIA in detail. We have made modifications in the method.
Here is the modified method:
GeneMANIA was used for analysis of Rabep1 co-expressed genes. We first choosed the human species study, then Rabep1 gene was directly put into the search box of GeneMANIA database to obtain the network diagram of genes co-expressed.
- In 2.5, Were rat PSCs used and the source? The description is confusing, “After transfection for 4 h, proteins were extracted 72 h after transfection” on line 119-120.
Reply: We are sorry that there is a misunderstanding in this statement and we have revised and added in the manuscript and highlighted it in red.
Here is the modified method:
“After transfection for 4 h, we replaced with conventional culture medium. Proteins were extracted 72 h after the medium replacement.”
PSCs:Rat primary pancreatic stellate cells (PSCs) were purchased from PUHE Biotechnology Company LTD (Wuxi, China) and were cultured with special culture medium (CTCC-102MCM) for rat primary PSCs in 37°C with 5% CO2.
- In 2.8, “standard deviation” should be abbreviated as “sd” or “SD”.
Reply: Thanks for your kind suggestion. We have modified it in manuscript.
Results:
- In Fig. 3 legend: Please define BL1-8 as human samples, CP or control. Were these 8 human samples different from 12 human samples mentioned in Methods?
Reply: We are sorry for the lack of this definition in our legend. We have made some modifications in the manuscript. BL1-5 represents the samples of patients with pancreatitis, and BL6-8 represents the normal samples.
The 12 samples mentioned in the method part included BL1-12, among which BL9-12 was the sample of idiopathic pancreatitis. We did not include this part of data in the research.
Discussion:
- “We speculate that trypsinogen is associated with the secretion of pancreatic acinar cells” on line 283-284. This should be a fact, not a speculation.
Reply: We are sorry that the description here is not precise and we have modified it in the manuscript.
Here is the modified version:
We have known that trypsinogen is associated with the secretion of pancreatic acinar cells.
- “Nafamostate is a serine protease inhibitor… Bafilomycin-A1 is a vacuolar H+-ATPase inhibitor…” on line 316-319. This is unnecessary repeat of the description in 2.1. Materials.
Reply: Sorry for the unnecessary duplication here, we have deleted part of the content.
Overall comments:
- Abbreviations: should be spelled out completely on initial appearance in text, for example, RAB GTPase Binding Effector Protein 1 (Rabep1).
Reply:
RAB GTPase Binding Effector Protein 1 (Rabep1), line 63
RAB5A, member RAS oncogene family (Rab5), line 68
α-smooth muscle agonin (α-SMA), line 355
- Figure labels: Should be consistent, either as Fig. or fig.
Reply: Thanks for your advice. We have unified fig. in the article into Fig.
- “Overexpression of trypsinogen”, and “rescued” are not appropriate to the content.
Reply: Thank you for your suggestion. We have modified the manuscript.
Here is the modified version:
Through Western blot, we observed that when PSCs (transfected with siRNA-Rabep1) compared to PSCs (untreated) were cocultured with AR42J (stimulated by TLC), the over-expression of trypsinogen was significantly rescued.
- Missing ref as ().
Reply: We are sorry for the misunderstanding, the parentheses here can represent functions in R software, which we have deleted in the article.
- In Funding: “This research received no external funding” or “This research was funded by NAME OF FUNDER, grant number XXX” and “The APC was funded by XXX”. Check carefully that the details given are accurate and use the standard spelling of funding agency names at https://search.crossref.org/funding. Any errors may affect your future funding” on line 351-354, is unnecessary.
Reply: Thanks for reminding me this mistake. We have turned the Acknowledgments content to Funding, and modified the Acknowledgments.
Reference:
- Sendler, M., Weiss, F. U., Golchert, J., Homuth, G., van den Brandt, C., Mahajan, U. M., Partecke, L. I., Döring, P., Gukovsky, I., Gukovskaya, A. S., Wagh, P. R., Lerch, M. M., & Mayerle, J. (2018). Cathepsin B-Mediated Activation of Trypsinogen in Endocytosing Macrophages Increases Severity of Pancreatitis in Mice. Gastroenterology, 154(3), 704–718.e10. https://doi.org/10.1053/j.gastro.2017.10.018
- Sherwood, M. W., Prior, I. A., Voronina, S. G., Barrow, S. L., Woodsmith, J. D., Gerasimenko, O. V., Petersen, O. H., & Tepikin, A. V. (2007). Activation of trypsinogen in large endocytic vacuoles of pancreatic acinar cells. Proceedings of the National Academy of Sciences of the United States of America, 104(13), 5674–5679. https://doi.org/10.1073/pnas.0700951104
- Ma, B., Wu, L., Lu, M., Gao, B., Qiao, X., Sun, B., Xue, D., & Zhang, W. (2013). Differentially expressed kinase genes associated with trypsinogen activation in rat pancreatic acinar cells treated with taurolithocholic acid 3-sulfate. Molecular medicine reports, 7(5), 1591–1596. https://doi.org/10.3892/mmr.2013.1355
- Shen, Y., Wen, L., Zhang, R., Wei, Z., Shi, N., Xiong, Q., Xia, Q., Xing, Z., Zeng, Z., Niu, H., & Huang, W. (2018). Dihydrodiosgenin protects against experimental acute pancreatitis and associated lung injury through mitochondrial protection and PI3Kγ/Akt inhibition. British journal of pharmacology, 175(10), 1621–1636. https://doi.org/10.1111/bph.14169
- Sendler, M., Weiss, F. U., Golchert, J., Homuth, G., van den Brandt, C., Mahajan, U. M., Partecke, L. I., Döring, P., Gukovsky, I., Gukovskaya, A. S., Wagh, P. R., Lerch, M. M., & Mayerle, J. (2018). Cathepsin B-Mediated Activation of Trypsinogen in Endocytosing Macrophages Increases Severity of Pancreatitis in Mice. Gastroenterology, 154(3), 704–718.e10. https://doi.org/10.1053/j.gastro.2017.10.018
- Zha, Z. Q., You, S., Hu, Y. H., Zhang, F., Chen, Y. W., & Wang, J. (2022). Asn57 N-glycosylation promotes the degradation of hemicellulose by β-1,3-1,4-glucanase from Rhizopus homothallicus. Environmental science and pollution research international, 10.1007/s11356-022-19959-5. Advance online publication. https://doi.org/10.1007/s11356-022-19959-5

Round 2
Reviewer 2 Report
Remaining issues:
Introduction:
1. A grammar error, “that internalization of PSCs in CP Activation of phagocytosis” on line 66-67.
I recommend: “…, suggesting activation of PSC endocytosis in CP.”
2. “We speculate that PSCs can induce endocytosis of trypsinogen by PSCs by regulating early endosome formation. Further trypsinogen was induced…” on line 72-74.
I recommend: “We speculate that PSCs can endocytose trypsinogen released from acinar cells, which involves endosome formation. The endocytosed trypsinogen can be activated in PSCs, aggravating pancreatic injury…”.
Materials and Methods:
1. In 2.2, I recommend to delete: “, and can be used to make acute pancreatitis cell and animal models by controlling drug concentration” on line 101-102.
2. In 2.3, line 133, “Microarry” should be “Microarray”; line 136 and 139, “1*PBS” should be “1xPBS”.
3. In 2.6, Please specify the application of GeneMANIA for the study.
I recommend: “GeneMANIA was used for analysis of Rabep1 co-expressed genes in human study (from what data base and what tissue/cell samples?”), and delete “We first choosed the human species study, then”.
Please clarify Figure 2 legend: Is (A) from rat cell study? Are (B), (C), and (D) from human data or samples?
4. In 2.10, “standard deviation” should be abbreviated as “sd” or “SD”, not (… ± s).
Overall comments:
1. Abbreviation in line 355, “α-smooth muscle agonin (α-SMA)” should be “…actin…”.
2. “Overexpression of trypsinogen”, and “rescued” are not appropriate to the content.
I recommend “Through Western blot, we observed that TLC treatment of AR42J cells increased trypsinogen levels in PSCs, while siRNA-Rabep1 transfection in PSCs reduced trypsinogen levels in PSCs.”
Author Response
Introduction:
- A grammar error, “that internalization of PSCs in CP Activation of phagocytosis” on line 66-67.
I recommend: “…, suggesting activation of PSC endocytosis in CP.”
Reply: We are sorry for the grammar error here, we have modified it as you suggested.
Revised as: “… , suggesting activation of PSC endocytosis in CP.”
- “We speculate that PSCs can induce endocytosis of trypsinogen by PSCs by regulating early endosome formation. Further trypsinogen was induced…” on line 72-74.
I recommend: “We speculate that PSCs can endocytose trypsinogen released from acinar cells, which involves endosome formation. The endocytosed trypsinogen can be activated in PSCs, aggravating pancreatic injury…”.
Reply: Thank you for your suggestion. We also think your modification is more logical and have modified it in the manuscript.
Revised as: We speculate that PSCs can endocytose trypsinogen released from acinar cells, which involves endosome formation. The endocytosed trypsinogen can be activated in PSCs, aggravating pancreatic injury…
Materials and Methods:
- In 2.2, I recommend to delete: “, and can be used to make acute pancreatitis cell and animal models by controlling drug concentration” on line 101-102.
Reply: Thanks for your suggestion, we have deleted this sentence.
- In 2.3, line 133, “Microarry” should be “Microarray”; line 136 and 139, “1*PBS” should be “1xPBS”.
Reply: Thank you for your careful and professional review comments. Under your guidance, we have found these problems and modified the manuscript.
- In 2.6, Please specify the application of GeneMANIA for the study.
I recommend: “GeneMANIA was used for analysis of Rabep1 co-expressed genes in human study (from what data base and what tissue/cell samples?”), and delete “We first choosed the human species study, then”.
Please clarify Figure 2 legend: Is (A) from rat cell study? Are (B), (C), and (D) from human data or samples?
Reply:
(1) We apologize for not explaining clearly how to use the GeneMANIA database. Sources of protein interaction information for this database are from public databases from a list of genes with their annotations and putative functions. These include various tissues or cells of multiple species and we have made changes in the manuscript.
Revised as: GeneMANIA was used for analysis of Rabep1 co-expressed genes in human study from many public databases such as (GEO, BioGRID, IRefIndex) and tissue or cell samples.
(2) We are sorry for not clarifying the Fig 2 legend. The data analysis of Fig2A, C and D were all derived from our Rat cell microarray data. The analysis in Fig2B was derived from an analysis of public databases of human tissues or cells. Considering that human samples are closer to the purpose of this study, protein interaction analysis of human samples is presented here. However, we also performed Rabep1 protein interaction analysis in rat samples, as shown below:
Figure 1 Analysis of Rabep1 co-expressed genes in rattus norvegicus through genemania database.
- In 2.10, “standard deviation” should be abbreviated as “sd” or “SD”, not (… ± s).
Reply: We are sorry for the omissions here. We have made modifications in the manuscript.
Revised as: All data are presented as the mean ± standard deviation ().
Overall comments:
- Abbreviation in line 355, “α-smooth muscle agonin (α-SMA)” should be “…actin…”.
Reply: We are sorry for the error here, we have made the correction in the manuscript.
Revised as: actin
- “Overexpression of trypsinogen”, and “rescued” are not appropriate to the content.
I recommend “Through Western blot, we observed that TLC treatment of AR42J cells increased trypsinogen levels in PSCs, while siRNA-Rabep1 transfection in PSCs reduced trypsinogen levels in PSCs.”
Reply: We are sorry for the inappropriate description here. We also think your description is more appropriate and have made modifications in the manuscript.
Revised as: Through Western blot, we observed that TLC treatment of AR42J cells increased trypsinogen levels in PSCs, while siRNA-Rabep1 transfection in PSCs reduced trypsinogen levels in PSCs.